# Drought identification in the Eastern Baltic region using NDVI

Egidijus Rimkus[1], Edvinas Stonevicius[1], Justinas Kilpys[1], Viktorija Maciulytė[1], Donatas Valiukas[2]

[1]Institute of GeoSciences, Vilnius University, Vilnius, 03101, Lithuania
[2]Lithuanian Hydrometeorological Service, Vilnius, 09300, Lithuania

*Correspondence to*: Egidijus Rimkus (egidijus.rimkus@gf.vu.lt)

**Abstract.** The droughts are the phenomena which affect large areas. Remote sensing data covering large territory can be used to assess the droughts' impact and their extent. Drought effect on vegetation was determined using Normalized Difference Vegetation Index (NDVI) and Vegetation Condition Index (VCI) in the east Baltic Sea region located between 53–60 °N and 20–30 °E. The effect of precipitation deficit on vegetation in arable land, broad–leaved and coniferous forest was analysed

using the Standardized Precipitation Index (SPI) calculated for 1 to 9 months time scales. Vegetation has strong seasonality in the analysed area. The beginning and the end of vegetation season depends on the distance to the Baltic Sea that affects temperature and precipitation patterns. The vegetation season in the south-eastern part of the region is 5–6 weeks longer than in the north-western part. The early spring air temperature, snowmelt water storage in the soil and precipitation has the largest influence on the NDVI values in the first half of the active growing season. Precipitation deficit in the first part of the

vegetation season has a significant impact only on the vegetation in the arable land. The vegetation in the forests is less sensitive to the moisture deficit. Correlation between VCI and the same month SPI1 is usually negative in the study area. It means that wetter conditions lead to the lower VCI values, while the correlation is usually positive between the VCI and previous month SPI. With longer SPI scale the correlation gradually shifts towards the positive coefficients. The positive correlation between 3 and 6 months SPI and VCI was observed in the arable land and both types of forests in the second half of vegetation season.

The precipitation deficit is only one of the vegetation condition drivers and NDVI cannot be used universally to identify droughts, but it may be applied to better assess the effect of droughts on vegetation in the eastern Baltic Sea region.

*Keywords:* Baltic Sea region, Standardized Precipitation Index, Normalized Difference Vegetation Index, Vegetation Condition Index, drought, land cover

## 1 Introduction

Vegetation indices derived from the remote sensing data are very important for the accurate assessment of the plant grow conditions, especially in the case of extreme weather events, such as droughts. Ground-based meteorological and agro–

meteorological drought indices only allow to evaluate the risks for agricultural lands, while the satellite information makes possible identification of damaged vegetation in various land types and assess the magnitude of damage.

Remote sensing of the vegetation condition is based on the fact that healthy plants have more chlorophyll and therefore absorbs more visible and reflects more near-infrared radiation (Myeni et al., 1995). Often vegetation conditions are determined by calculating the Normalized Difference Vegetation Index (NDVI). Since 1981 this index is provided on a global scale using

Advanced Very High Resolution Radiometer (AVHRR) on–board of NOAA satellites.

The long-term data set is a big advantage, but the problems may arise in interpreting the index changes. During more than 30 years of measurements, the land use has been changed in many locations and it is difficult to determine the climatic signal in the NDVI changes. The accuracy of the growing conditions evaluation depends on the assessment of environmental and atmospheric conditions as well as peculiarities of the sensor response (Jackson and Huete, 1991). During the period without

precipitation, NDVI values can decrease not only due to the deterioration of the plant but also due to the increase of dust in the air and on the surface of the plant, which is usually "washed out" along with the rain. For this reason, the vegetation index can have lower values than they should (Mirzaei et al., 2011).

It is necessary to emphasize that the vegetation (and hence NDVI values) response to the meteorological conditions in a given year depends on the geographical region and environmental factors such as vegetation type, soil type and land use (Usman et

al., 2013). NDVI is a good indicator of vegetation–soil moisture conditions, but seasonality should be taken into account when using this index for drought monitoring (Ji and Peters, 2003). Therefore, in most cases NDVI values are analysed in complex with ground-based meteorological and agro–meteorological drought indicators such as Standardized Precipitation Index (SPI) (Ji and Peters, 2003; Bhuiyan et al., 2006; Quiring and Ganesh, 2010; Gebrehiwot et al., 2011; Gaikwad and Bhosale, 2014; Stagge et al., 2015), Standardized Precipitation–Evapotranspiration Index (SPEI) (Stagge et al., 2015), standardized Water–

Level Index (SWI) (Bhuiyan et al., 2006), Palmer Drought Severity Index (PDSI), Moisture Anomaly Index (z–index) (Quiring and Ganesh, 2010). The most commonly the spatial and temporal variability of the drought is associated with precipitation deficit, so SPI index is often used due to its simplicity (Gebrehiwot et al., 2011).

Previous studies have shown that the NDVI and SPI values are correlated and this relation is the strongest in the middle of the active growing season, and the weakest at the beginning and at the end (Ji and Peters, 2003). However, not in all cases negative

NDVI anomaly can be related with low SPI values (Bhuiyan et al., 2006). The strongest relationship between SPI and NDVI was found in the areas with low soil water–holding capacity (Ji and Peters, 2003). Also, the relationship between these two indices differ in various agricultural areas: a positive correlation between SPI and NDVI was determined in the rain–fed areas, while negative in the irrigated areas (Ozelkan et al., 2016). The SPI indicates moisture conditions and the vegetation reacts to the lack of precipitations with some delay. For this reason, the strongest link was established between the SPI values in spring

and NDVI values in summer, which means that spring watering is critically important for the growth of the most plants (Ozelkan et al., 2016).

Frequently NDVI index is analysed by calculating Vegetation Condition Index (VCI), which compares the current NDVI to the observed values of this index in previous years (Gebrehiwot et al., 2011; Ozelkan et al., 2016) and have a good correlation

with the SPI values (Dutta et al., 2015). In different regions of the world, the relationship between the three (Gebrehiwot et al., 2011), six or nine months SPI (Quiring and Ganesh, 2010) and VCI values were established. Some studies showed that the impact of the short–term precipitation fluctuations on VCI values is weak (Quiring and Ganesh, 2010).

The eastern coast of the Baltic Sea is in a transitional area from the maritime to the continental climate, characterized by a strong west-east gradient in the continentality of climate (Jaagus et al., 2010; Jaagus et al. 2014). The region can be divided into three geographical zones: the western coastal areas, the central agricultural zone with fertile soils, and the eastern region which contain most lakes, swamps, and forest. The spatial pattern of seasonal temperature and precipitation in the Baltic countries depends on the two main large-scale factors: latitude and the Baltic Sea (Jaagus et al., 2010; Jaagus et al. 2014). In the eastern Baltic region droughts are not a very common (Loyd-Hughes and Saunders, 2002), but they can cause significant losses for economy and wildlife (BACC, 2008). During the second half of the 20th century the dryness of the region remain similar or even decreased (Bordi et al., 2009; Rimkus et al., 2012; Rimkus et al., 2013), but recent studies show that in the 21st century, at least in some parts of the study area, the water availability is likely to decrease in summer and autumn (Stonevicius et al., 2017). Dry periods are related to the atmospheric circulation patterns and can affect large areas (Rimkus et al., 2014). In temperate and boreal climate the effect of water shortage on vegetation is not as significant as in arid or semiarid areas, but there is strong evidence of water deficit effect on various types of vegetation in the study area (Kulikauskas and Sprainaitienė, 2005; Vitas and Erlickyte, 2008; Ozolincius et al., 2009; Kalbarczyk, 2010; Bijak, 2011).

The NDVI and VCI have not been used for the drought analysis in the Baltic Sea region yet. In this region, especially in the southern part, the agriculture is strongly developed, and development of the new methods for evaluation of drought extension and intensity are very important.

The main objectives of this study are to determine the impact of the droughts on the plant active growing conditions in the eastern part of the Baltic region, to identify other factors which may lead to negative NDVI anomalies or low VCI values during the active growing season, and to find links between SPI and VCI values. Areas of arable land, broad-leaved and coniferous forests were analysed separately and differences of precipitation deficit impact on vegetation in different land use types were determined.

## 2 Data and methods

The analysed area covers the eastern part of the Baltic Sea region and is located between 53 to 60 °N and 20 to 30 °E (Fig. 1). The NDVI index was used to analyse the vegetation condition in the land cells of study area. CORINE (Coordination of Information on the Environment) land cover data was used to identify the response of vegetation in different land use types to precipitation deficit. CORINE land cover data is available only in part of study area covering Estonia, Latvia, Lithuania and Poland (Fig. 1).

The NDVI data set was obtained from NOAA STAR–NESDIS system which generates global and regional vegetation health data. The NDVI index is derived from the radiance observed by the Advanced Very High Resolution Radiometer (AVHRR)

on–board of polar orbiting satellites: the NOAA–7, 9, 11, 14, 16, 18 and 19. NDVI is calculated as the difference between reflectance in near-infrared (NIR) and visible red (VIS) by following Eq. (1):

$$NDVI = \frac{(NIR - VIS)}{(NIR + VIS)} \tag{1}$$

The NOAA STAR–NESDIS NDVI product has 16 km spatial and 7–day composite temporal resolution and covers a period from 1981. In this study data from 1982 to 2014 was analyzed. NDVI data set is generated using maximum–value composite (MVC) method (Holben, 1986). This method reduces the influence on NDVI from clouds, spectral properties, resolution, and residual atmospheric effects that all act to reduce NDVI (Scheftic et al., 2014). NOAA STAR–NESDIS system produces no noise NDVI. The NDVI is filtered in order to eliminate the high-frequency noise. It is also adjusted for a non–uniformity of the land surface due to the climate and ecosystem differences using multi–year NDVI and brightness temperature data. Final NDVI product is provided in the geographic grid with equal latitude and longitude interval (0,144°×0,144°) (NOAA NESDIS, 2013). The data set has several gaps: from 50 week of 1984 to 8 week of 1985; from 37 week of 1994 to 3 week of 1995; from 2 to 4 week, from 11 to 24 week and 29 week of 2004.

The NDVI values range from -1 to +1. The negative index value can be recorded over the water bodies while values are close to 0 over the land without vegetation. The index value equal to 1 indicates perfect growing conditions (Lillesand and Kiefer, 1994; Belal et al., 2014).

The total number of analysed cells is equal to 2184. 31 cells in the coastal areas or near the big lakes were unequally recognized as a land or sea cells by different satellites, and in some cases, the information was missing. In such cases, data derived from the particular cells were excluded from further analysis. Also, 99 cells near the sea coast and probably partly covered by the sea, had negative NDVI values during the active growing season and thus were excluded from analysis. In total 6 % of the initial data set was not used in the study.

The NDVI values are influenced not only by the natural variation and health of vegetation. In long–term data sets, variability related to the satellite orbital drift, sensor degradation, and satellite change are also determined (Kogan, 1997). The initial trend observed in this research was mostly related to the satellite change and this trend was removed by applying a systematic correction for each satellite data separately.

Active plant vegetation in the eastern Baltic region starts when the daily mean air temperature exceeds 10 °C. In majority of years it happens in the first half of May. End of the active growing season usually occurs in the second half of September. It is necessary to mention that there are up to several weeks differences in growing season in the sudy area (due to the latitude and distance from the sea), as well as quite large year-to-year variation of such dates.

Since the drought makes the greatest impact on the plants during the active growing season data from 18–39 weeks of the year (May–September) were analysed. Not only absolute NDVI values were evaluated, but also their deviations from the mean. For this reason, Vegetation Condition Index (VCI) (Kogan, 1995) was calculated. VCI compares the current NDVI with measured historical NDVI values. It is defined as following Eq. (2):

$$VCI = \frac{NDVI - NDVI_{min}}{NDVI_{max} - NDVI_{min}} \times 100, \tag{2}$$

where NDVI – measured monthly (weekly) value, $NDVI_{min}$ and $NDVI_{max}$ – historical minimum and maximum values of analysed month (week). Lower VCI values indicate bad, while higher values show a good vegetation state. The VCI is expressed in % and vary from 0 to 100 and according to Kogan (2002) low values below <40 can be described as mild drought, < 30 as moderate drought, <20 as severe drought and < 10 as extreme drought.

According to Jain et al. (2010), the VCI is a better indicator of the moisture deficit than NDVI because it allows to separate the short–term climate signal from the long–term ecological signal. VCI enables to compare simultaneously measured NDVI values not only under the different geographic conditions but also in the different vegetation types.

VCI vary from 0 to 100 and according to Kogan (2002) low values below <40 can be described as mild drought, < 30 as moderate drought, <20 as severe drought and < 10 as extreme drought.

CORINE land cover data was used to identify the dominant land use type in the NDVI cells (0,144°×0,144°). CORINE data sets with 100 m resolution were used. The CORINE data sets with reference years 1990 (CLC 1990) and 2012 (CLC 2012) were compared to identify the land use changes during the study period. It was considered that land use in particular NDVI cell was stable if CORINE land use classes coincided at least in the 80 % of the cell area. From the set of cells with the stable land use the cells with different dominant land use types were identified. The diverse land use is common in analysed region. To reduce the number of CORINE land use classes the mixed forests and transitional woodland–shrub areas were joined with broad–leaved forests and broad–leaved vegetation class was formed. On average, the broad–leaved vegetation class consisted of 28 % of broad–leaved forests, 57% of mixed forests and 15 % of transitional woodland–shrub areas. It was considered that the land use class is dominant if it covers at least 50 % of the cell area. Only three types of land uses were identified as dominant in more than 5 cells: arable land (209 cells), broad–leaved (80) and coniferous forest (25 cells) (Fig. 1). These 3 land use types were used in this study to differentiate the effect of climatic conditions on vegetation.

Four cases with strong NDVI anomalies were investigated. Winter 1987 was one of the coldest during the entire study period in the whole eastern Baltic region. Also, it was the only year when the mean March–April temperature averaged over the entire area was negative (-1 °C), and it has led to a very late beginning of the vegetation season. In 1990, after one of the warmest winters, the particularly high air temperature in March–April was recorded (5.9 °C), and this has led to a very early beginning of vegetation. Years 1992 and 2002 were analysed because the largest negative precipitation anomalies in May–September were recorded, respectively 37 and 43 % below long–term average. During these years the agro–meteorological droughts have been observed in the substantial part of the analysed area (Valiukas, 2015).

In order to assess the impact of precipitation deficit on vegetation condition, the Standardized Precipitation Index (SPI) was used in this study. The SPI calculation for any location is based on a monthly rainfall data series, first applying gamma distribution and then transforming it into a normal distribution (McKee et al., 1993; Edwards and McKee, 1997). Positive SPI values indicate greater than average precipitation amounts while negative values indicate lower amounts (Table 1).

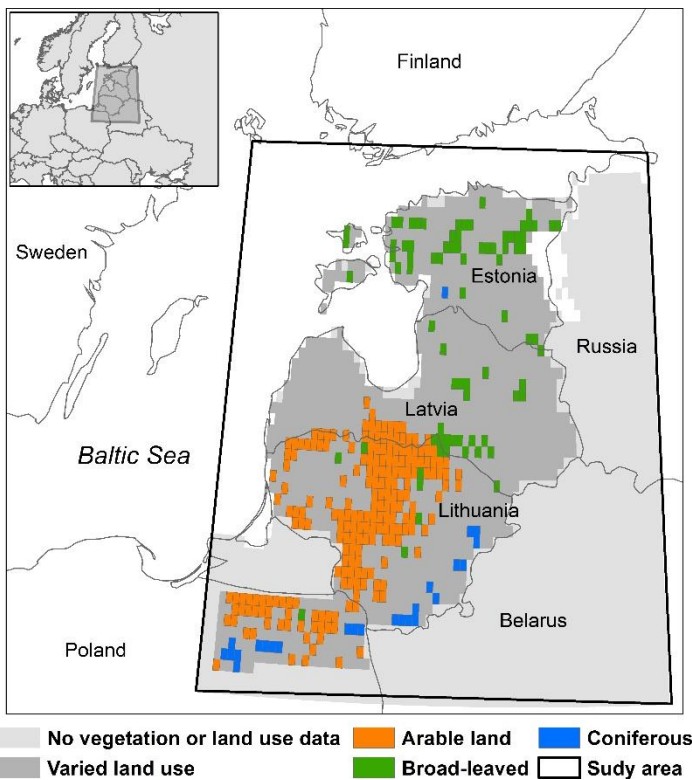

**Figure 1: Study area in the eastern part of the Baltic Sea region. The dominant type of land use in 0,144°×0,144° cells was estimated according to the CORINE land use data.**


**Table 1: Interpretation of SPI values (McKee et al., 1993).**

| Value | Interpretation |
|---|---|
| ≥2,0 | Extremely wet |
| 1,99– 1,5 | Very wet |
| 0,99 – -0,99 | Near normal |
| -1 – -1,49 | Moderately dry |
| -1,5 – -1,99 | Severely dry |
| ≤-2,0 | Extremely dry |

High–resolution (0.5°×0.5° latitude/longitude) monthly precipitation data from CRU TS (Climate Research Unit Time Series) data set (Harris et al., 2014) has been used in this study to calculate SPI values. The initial analysis indicated that the vegetation conditions are not strongly affected by the moisture deficit calculated for the time scales above 9 months. The effect of long-
term SPI might be weakened by the conditions of the cold season when water supply depends on the precipitation type,

snowmelt and soil condition. The 1–, 3–, 6– and 9–month SPI values (SPI1, SPI3, SPI6) were used in this study to investigate the effect of short and medium term precipitation deficit.

## 3 Results

### 3.1 Spatial and temporal variation of NDVI

Vegetation has a very strong seasonality in the Eastern Baltic region due to the variation of the day length, insolation and air temperature. During the cold season NDVI values in most of the cells are below 0.1 and begin to increase in the second half of March (Fig. 2). The NDVI change from March till May follows a clear spatial pattern. Firstly the NDVI increases in the southern part of the study area and near the Baltic Sea coast. With time vegetation index increases towards the northeast. At the end of April NDVI exceeds 0.2 in all study area. The largest NDVI values are reached in June and July. A peak of vegetation

was usually recorded on June 18-24. Smaller values (NDVI<0.50) are more common in the southern part of the domain (Fig. 2). NDVI begins to decline in August. In September NDVI in a majority of the study area drops below 0.40. Since the beginning of October, the NDVI values starts to decrease from the north-eastern part of the study area and in the beginning of November NDVI values remain larger than 0.2 only in the several cells located in the south-western part of the domain (Fig. 2). The length of the period with NDVI higher than 0.2 in the south-western part of the study area is 5–6 weeks longer than in the

north-eastern part.

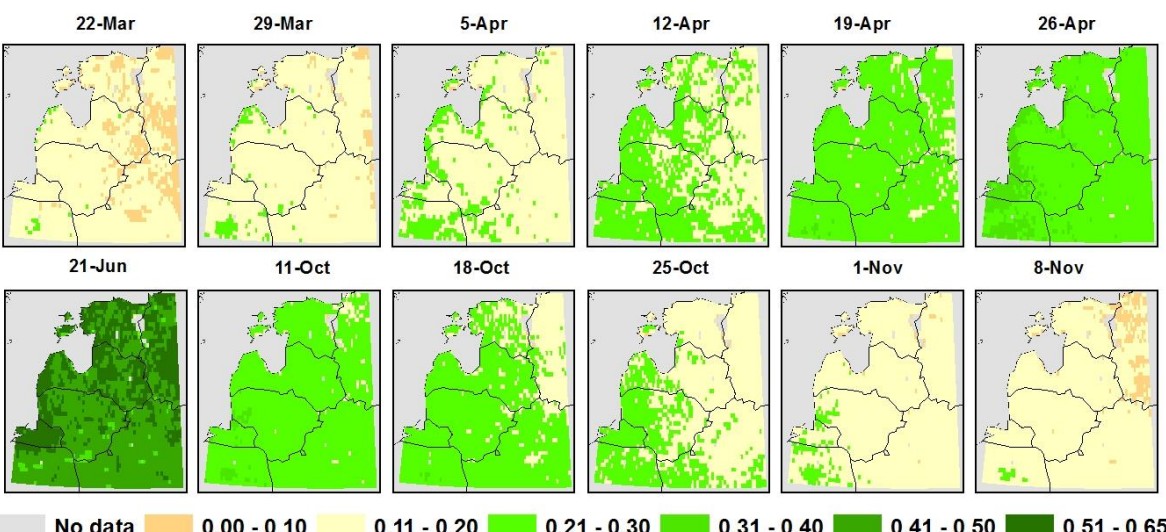

**Figure 2: Median of weekly 1982–2014 NDVI in spring (from March 22 to April 26), autumn (from October 11 to November 8) and during the vegetation peak (June 18-24) in the eastern part of the Baltic Sea region. Dates are the mid-points of weekly NDVI data.**

There is a clear difference in the seasonal pattern of NDVI in the different land uses (Fig. 3). NDVI in the cells with dominant arable land cover and broad-leaved vegetation is below 0.20 until the middle of April. Later it gradually increases till the first

half of June. From the second half of June, NDVI decreases in the cells with both arable land and broad-leaved vegetation. The rate of NDVI decreases in the cells with arable land is much sharper than the one in the cells with broad-leaved vegetation. The difference in the NDVI pattern may be attributed to the difference in vegetation type and land management practices. The

annual plants are commonly seeded in the arable land and such type of vegetation has a faster vegetation cycle. On the other hand, the crops in the case study area are harvested in the August and September. NDVI has lower seasonality in the cells dominated by the coniferous vegetation. In this land cover type the NDVI values remain higher than in other vegetation classes during the cold season and are more stable during the warm season, but on average the highest NDVI values do not exceed 0.4 (Fig. 3).


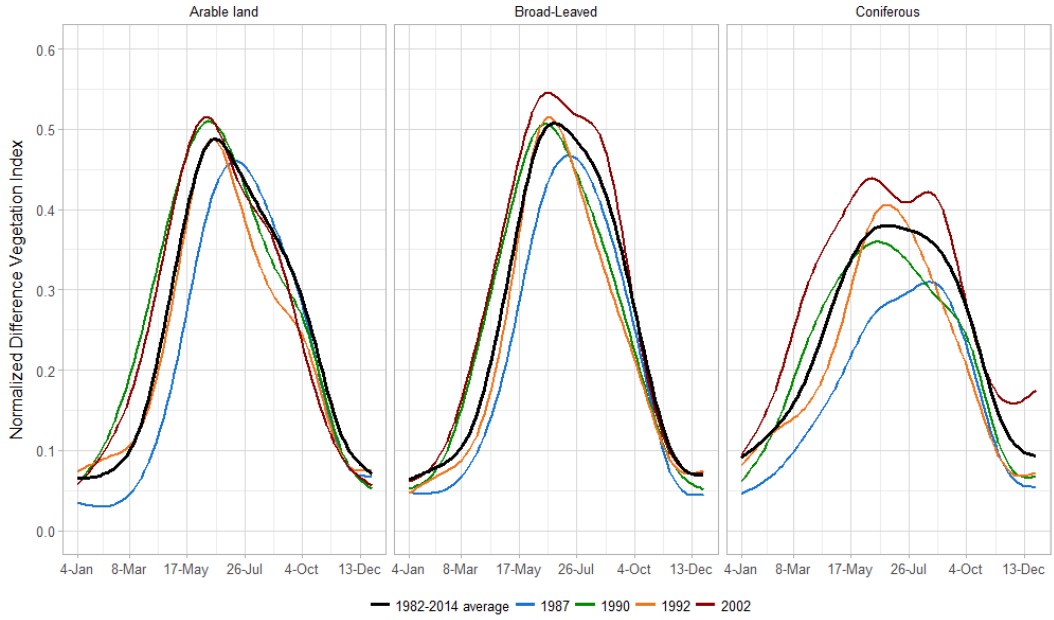

**Figure 3: NDVI profile for different land uses of multi–annual average (1982–2014), years with cold (1987), warm (1990) winter and spring seasons, and during the years with precipitation deficit (1992, 2002).**

During the particular year, the seasonal NDVI pattern may considerably differ from the multi–annual average. In 1990 and

2002 spring was warmer than usual and it's likely led to the higher NDVI values in the first half of the active growing season in the cells with all land uses (Fig. 3). The 2002 summer was among the driest during the analysed period, but despite that the NDVI remained higher than average in the arable land until the end of July and in the cells with broad-leaved and coniferous vegetation the NDVI remained above the average till the beginning of October. In 1987 winter and spring were colder than usual. It led to the 2–3 weeks later start of the vegetation season. In all land use classes, the NDVI was considerably smaller

than average. The late start of vegetation in 1987 also led to the late end but not in all years there is a close positive correlation between these dates (Fig. 3).

**3.2 Vegetation condition during the years with different hydrothermal regime**

NDVI has a strong seasonal pattern. VCI index compares the current NDVI to the range of values observed in the same period in previous years (Eq. 2) and is more suitable to illustrate the deviation of vegetation condition from normal (Jain et al., 2010).

If the VCI is lower than 50, the vegetation conditions are worse than normal. VCI values in May and June following the cold spring (1987) are smaller than 20 in the majority of the study area (Fig. 4). Within a few months, the vegetation reaches normal condition again. Warm spring of 1990 led to the better vegetation condition in May and June. In the large part of the area, VCI values were higher than 80. Intensive vegetation in the first part of the year gradually turned into low VCI values in the second half of vegetation season (Fig. 4).


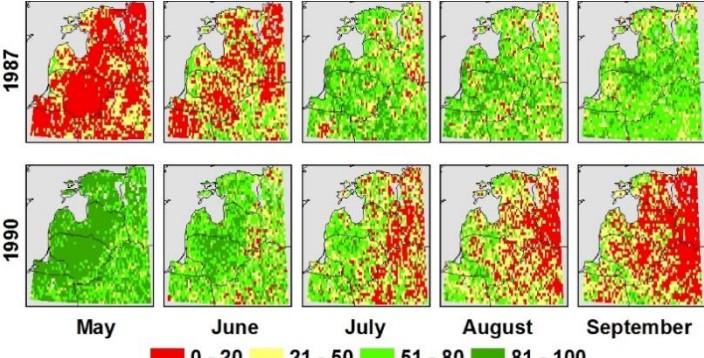

**Figure 4: Vegetation condition index (VCI) during the year with cold winter and spring (1987) and warm winter and spring (1990).**

It seems that the precipitation deficit might not be the decisive factor determining the vegetation condition in the eastern Baltic

Sea region. Years 1992 and 2002 had a lower than normal precipitation amount during the vegetation season, but in 1992 the vegetation was affected much more than in 2002 (Fig. 5). June 1992 was particularly dry. SPI1 representing one-month precipitation deviation from the norm was lower than -2.0 in the large part of the study area (Fig. 5a). July was dry only in the south-eastern part of the analysed region. Both SPI3 and SPI6 which represent the dryness for 3 and 6 months respectively were the lowest in July and August. Vegetation condition started to decline in some cells in July but in the majority of the

study area the VCI felt below 20 in August and remained similar in September.

The vegetation season of 2002 was also exceptionally dry (Fig. 5b). In July the precipitation deficit was observed in the eastern part of the region, while in August extreme meteorological drought (SPI1≤-2.0) was determined almost in all area except the south-eastern part. The precipitation deficit at the beginning of vegetation season was small but it gradually accumulated with time and in August and September SPI3 and SPI6 indicated severely or extremely dry conditions in the large part of the study

area. However, vegetation was affected only in the southern part of the region (Fig. 5b).

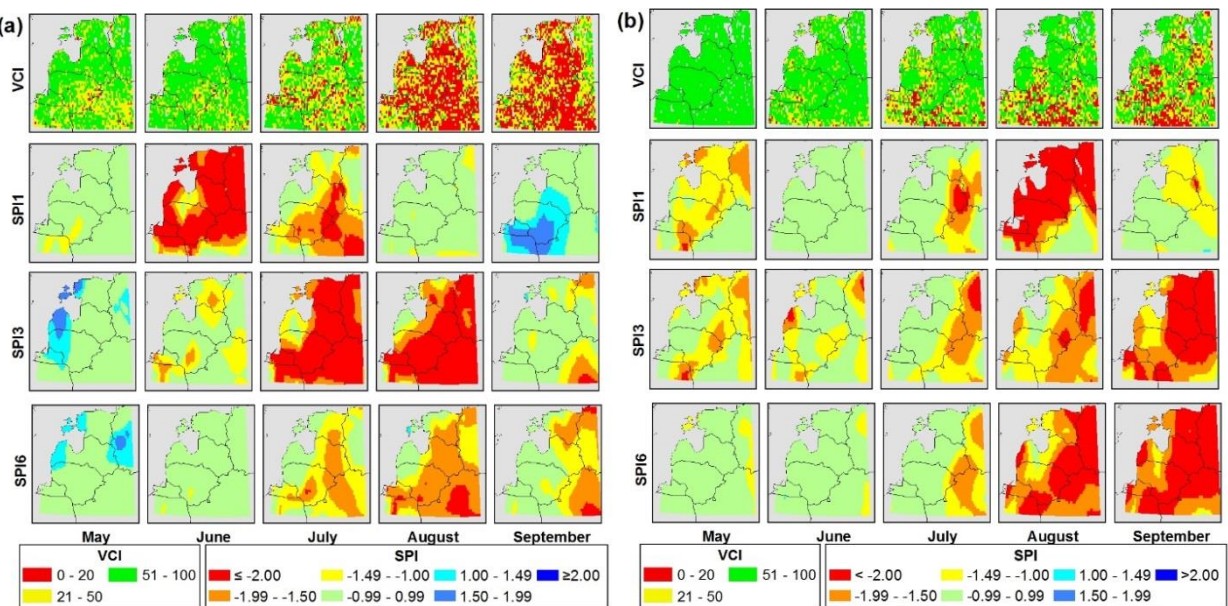

**Figure 5: Vegetation condition index (VCI) and standardized precipitation index (SPI) during the dry years of 1992 (a) and 2002 (b).**

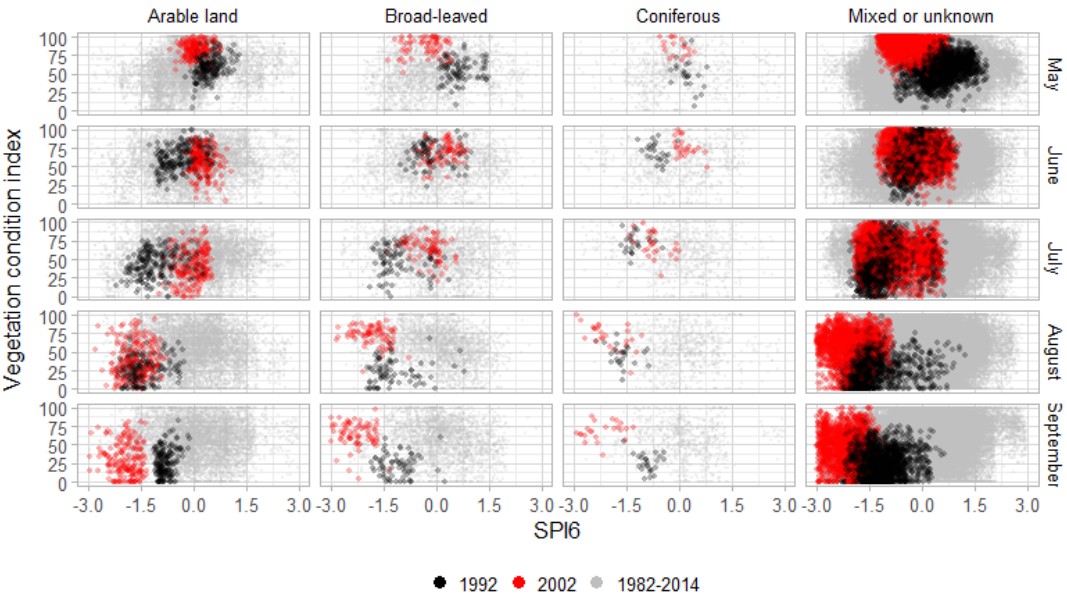


**Figure 6: Relationship between SPI6 and VCI for different land use types during the normal (1982–2014) and dry (1992, 2002) years.**

The most important distinction between dry 1992 and 2002 years was the reaction of vegetation in different land use classes to the precipitation deficit (Fig. 6). In both cases, the vegetation in the arable land was in good condition in May. Since June

VCI values in the arable land decreased and in August–September there were a lot of cells with VCI<20. In land use classes with the broad–leaved and coniferous vegetation the reaction to the precipitation deficit was different during 1992 and 2002, e.g. VCI has decreased more significantly in 1992 than in 2002. On the other hand, one, three and six-month SPI values calculated for August and September were lower in 2002 than in 1992.

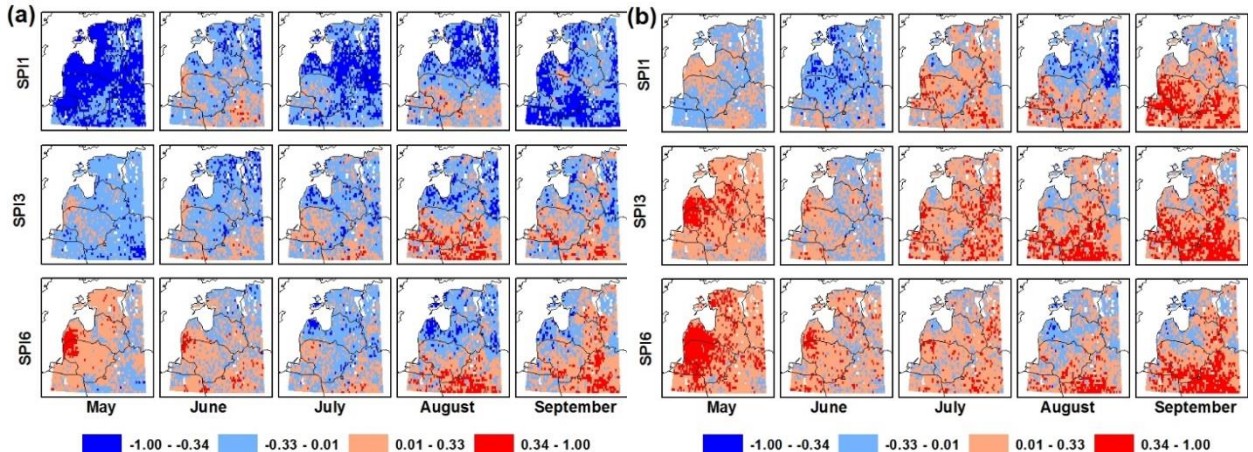


**Figure 7: Pearson correlation coefficient between VCI and the same month SPI values (a) and between VCI and SPI with one month lead (b). Coefficients large than 0.34 and smaller than -0.34 are statistically significant at 0.95%.**

Pearson correlation coefficient between monthly VCI and SPI was calculated to identify the effect of precipitation deficit on the vegetation in a particular cell. The correlation between VCI and the same month SPI1 is usually negative in the study area

(Fig. 7a). The negative correlation coefficient shows that higher SPI or wetter conditions lead to the lower VCI values. With longer SPI scale the correlation gradually shifts towards the positive coefficients (Fig. 7a). There is a weak spatial pattern of correlation coefficient distribution. The coefficients in the northern part of the study area tend to be negative while in the southern part correlation in the most cells is positive (Fig. 7a). The correlation is usually positive between the VCI and previous month SPI (Fig. 7b). There is a cluster of cells with a statistically positive correlation between VCI in May and SPI in April in

the western part of the study area. In August and September, a statistically significant positive correlation is common in the southern part of the region. The pattern of correlation between SPI and VCI implies the existence of spatial factor affecting the relationship. This pattern of VCI and drought indexes has been observed in other studies as well (Quiring and Ganesh, 2010).

When vegetation is affected by a certain factor its condition may remain distressed for some time. Only the months during

which the VCI for the first time dropped below 20 were used to identify how the SPI values are distributed when VCI indicates poor vegetation condition (VCI<20) (Fig. 8). The distributions of SPI values one month before VCI drops below 20 have weak positive skew. A higher density of SPI values indicating severely or extremely dry conditions (SPI≤-1.5) could be expected if the decrease of vegetation condition would be caused mainly by the significant precipitation deficit. In majority of the months

and all land use classes the SPI values indicated normal or moderately dry conditions (-1.49<SPI<1.0) before VCI dropped
below 20. However, according to SPI1 the vegetation condition in June can even worsen after wet May (Fig. 8).

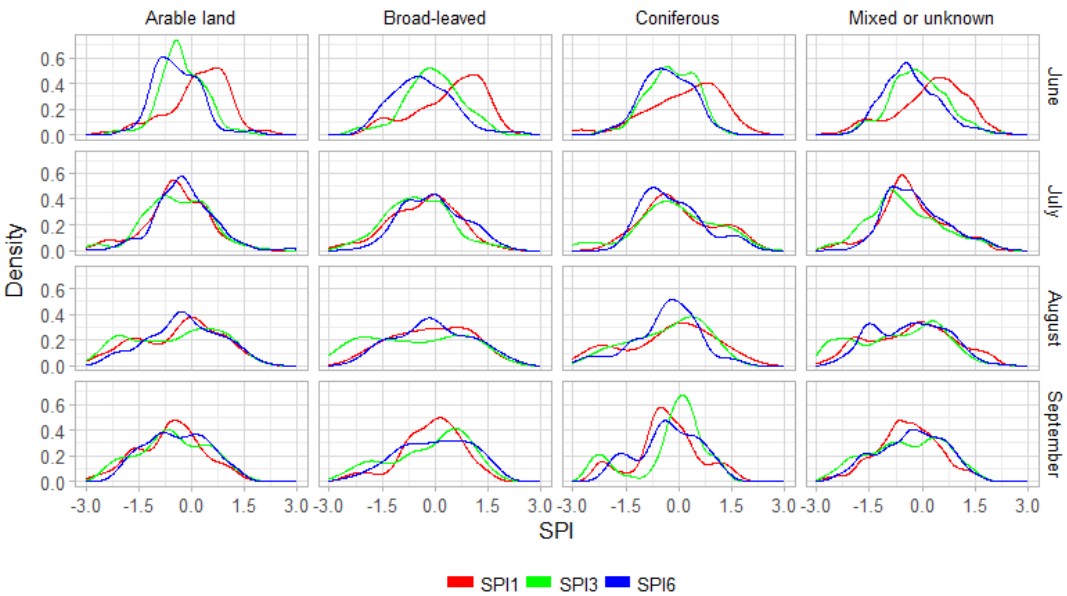

**Figure 8: The distribution of SPI values for different land uses one month before VCI drops below 20.**

**4 Discussion**

NDVI values in the analysed area are determined by a number of climatic factors. On the average, the active growing season
in the Baltic States lasts from the end of April until the beginning of October. The spatial pattern of the seasonal NDVI variation
is closely related to the distance from the sea, because the Baltic Sea is a major factor, determining the temperature and
precipitation regime in the analysed area (Jaagus et al., 2010; Jaagus et al., 2014). The differences in NDVI values in spring
and autumn in the west–east direction are larger than in the south–north. The south–north NDVI gradient would be more
noticeable if the determining factors would be the day length and insolation.

Many studies show, that beginning of the active growing season is determined by the spring temperature prior the event (Jeong
et al., 2011; Shen et al., 2014) in the temperate and high latitudes of the Northern hemisphere. In spring the soil is saturated
with melting snow water and excess moisture can worsen vegetation condition. Agricultural activity in the arable land usually
starts when soil becomes rather dry. For this reason, in the case of abnormally wet spring, the negative NDVI anomalies can
be recorded in May, which may be associated with the crop area and not to the crop conditions (Zhang et al., 2014). On the
other hand, in June moisture deficit in the arable soil may lead to vegetation deterioration, while this effect in conifer and

broad–leaved forests is almost invisible. Such differences can be explained by the fact that in the arable land mostly annual crops with relatively shallow roots are grown, so the lack of moisture may occur even during the short dry period. The Northern hemisphere boreal forests usually grow in the areas of excessive moisture, tree roots are deeper, so they react much slower to the precipitation deficit (such deficit may even lead to the higher VCI values). Also, due to the high initial soil moisture, the drought impact on forested areas can be minimal (Gao et al., 2016). Only during the prolonged extreme droughts (e.g., 1992), the VCI values in the forests of the study area decreased significantly.

Many studies indicate that VCI reacts with a delay to the change of moisture conditions and this reaction is controlled by the previously accumulated soil water storage (Quiring and Ganesh, 2010). Therefore, the strongest connection between SPI and VCI was determined in the areas with low soil water–holding capacity (Ji and Peters, 2003). Other studies also showed that forests respond to the drought on the long-term scales, while arable land on the short-term scales (Li and Zhou, 2015).

In the second half of the active growing season, the positive correlation between SPI3, SPI6, and VCI has been determined in the large part of the territory. A positive and in many places statistically significant correlation was found in the first half of the season if SPI with one month lead was used. However, the moisture deficit has a significant impact on vegetation condition only in the second half of the active growing season in the analysed part of the Baltic Sea region. Meanwhile, in the drier areas, SPI6 and SPI9 has a strong positive correlation with VCI throughout the year (Quiring and Ganesh, 2010).

It should be mentioned that due to the early start of the active growing season the peak of vegetation usually is reached earlier, and early vegetation start not always leads to an increase in aboveground production (Livensperger et al., 2016). Therefore, the low VCI values in August and September are not always related to the precipitation deficit. However, the time when the NDVI values fall close to the typical winter values (end of October - a beginning of November) in arable soils and broad–leaved forests are very similar during the years with the different hydrothermal regime (Fig. 3). This is related to the routine agricultural practices on the arable lands (when the land is plowed in autumn) while the fall of tree leaves is associated with the occurrence of the first intense frosts, which are usually in October. Meanwhile, in the coniferous forests, the differences of NDVI values that form in the summer months remain until the end of the calendar year (Fig. 3).

In the arid and semiarid areas, the spatial and temporal patterns of vegetation are primarily related to precipitation (Wang et al., 2001). The analysed region has a surplus precipitation during the most of the years, thus the air temperature anomalies might be the limiting factor for vegetation condition, especially in the transitional seasons (spring and autumn). The comparison of 1987 (cold spring) and 1990 (warm spring) also indicates that the interpretation of VCI values as an indicator of drought severity (Kogan, 2002) may not be universal in the eastern part of the Baltic Sea region. Dabrowska–Zielinska et al. (2002) also found that in the nearby Poland the VCI plays the minor role in defining of vegetation condition and crop yield.

**Conclusions**

The early spring air temperature, snowmelt water storage in the soil and precipitation has the largest influence on NDVI values in the first half of the active growing season. Negative correlation between SPI1 and VCI shows that the short–term precipitation deficit leads to the better vegetation condition. With longer SPI scale the correlation coefficients become positive

which shows that longer periods of moisture deficit reduces vegetation condition. The correlation is usually positive between the VCI and previous month SPI because vegetation responds to the external forcing with a delay.

The precipitation deficit in the first part of the vegetation season has a significant impact only on the vegetation in the arable land, while vegetation in the forests is less sensitive to the moisture deficit in the first half of the active growing season. The

positive correlation between 3- and 6-months SPI and VCI was observed in the arable land and both types of forests in the second half of the vegetation season.

The precipitation deficit is only one of the vegetation condition drivers in the eastern Baltic region and NDVI or VCI cannot be universally used to identify droughts, but it may be applied to better assess the effect of droughts on vegetation and the crop damage.

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
