# Peer review of "Drought identification in the Eastern Baltic region using NDVI"

_Earth System Dynamics, 2017_

## Referee Comment (RC1) · Anonymous Referee #1 · 7 Apr 2017

In this article, relationships between the Normalized Difference Vegetation Index (NDVI) measured from satellites and drought in the Eastern Baltic region. The topic is interesting and of practical importance. The paper is well prepared and I have no large critical problems. But I have a number of specific remarks for correcting and improving the text. I'll answer to the general questions of the journal and then I'll make my more detail comments and suggestions. 1. Does the paper address relevant scientific questions within the scope of ESD? Yes. 2. Does the paper present novel concepts, ideas, tools, or data? Yes. 3. Are substantial conclusions reached? Yes. 4. Are the scientific methods and assumptions valid and clearly outlined? Yes. 5. Are the results sufficient to support the interpretations and conclusions? Yes. 6. Is the description of experiments and calculations sufficiently complete and precise to allow their reproduction by fellow scientists (traceability of results)? Yes. 7. Do the authors give proper

credit to related work and clearly indicate their own new/original contribution? Yes. 8. Does the title clearly reflect the contents of the paper? Partly. There is mentioned only NDVI but not VCI. 9. Does the abstract provide a concise and complete summary? Partly. Results from the section 3.2 were not presented. 10. Is the overall presentation well structured and clear? The use of the VCI in not well presented in the abstract, introduction and methods. 11. Is the language fluent and precise? Could be improved. 12.Are mathematical formulae, symbols, abbreviations, and units correctly defined and used? Yes. 13. Should any parts of the paper (text, formulae, figures, tables) be clarified, reduced, combined, or eliminated? No. 14. Are the number and quality of references appropriate? Yes. 15. Is the amount and quality of supplementary material appropriate? Yes.

Remarks and suggestions 1. The abstract is too general. I would like to see more concrete results of the study in the abstract. 2. Page 1 line 29. There is written "Remote sensing of the vegetation condition is based on the fact that healthy plants have more chlorophyll and therefore absorbs more visible and reflects more infrared radiation (Myeni et al., 1995)". Plants reflect short-wave radiation and emit infrared radiation. I assume that here should be "... and emit more infrared radiation". 3. Page 2 line 39. I suggest that vegetation response here not to climatic impacts but to different meteorological conditions each year. 4. In fact, the NDVI describes the greenness of the underlying surface. Is the term "greenness" used in this context and could it be used in this article? 5. I suggest more extended overview of the literature on climatological studies of droughts in the study region in the introduction. 6. Page 3 line 71. I suggest more exact definition of the study area. It does not cover the whole territory between 53-60 N and 20-30 E. Could it be identified as Estonia, Latvia, Lithuania and northeastern Poland? There is a contradiction between Figure 1 and other maps. In the first case, only these countries are shown on the map while in the second case, all land areas in the domain have been used. Can you explain this difference in the definition of the study region? 7. Page 3 line 79. The end of the study period is not shown (from 1981 to the present). What is the last year? 8. Page

3 lines 85-86. Week numbers are usually not used in the everyday life. Therefore, it is not informative to used week numbers from the beginning of a year. The use of dates is more useful in my mind. 9. Page 4 Line 99. What is mean average daily air temperature? Is it simply daily mean temperature? 10. Page 4 pages 100-102. The beginning and the end of the active growing season varies significantly over the study region in dependence of latitude. It could be emphasise in the text. These dates are very variable also from year to year. 11. Figure 1. It is very strange that the territory of Estonia and major part of Latvia is described only by broad-leaved forest. There are no cells from coniferous forest and arable land. This selection of cells does not represent adequately the vegetation types in that region. It should be explained somehow why these cells were selected. Here are also problems related to the CORINE land cover data. In the figure caption the CORINE data should be mentioned. 12. It is not clear how the VCI values were calculated in this study. It should be described because they are used in this study. 13. Page 6 line 152. It is interesting to know when the week 25 takes place. I am not sure that the large values are more common in the northern part of the domain (Figure 2). Lower values are in the southern part but in the other regions there are not such territorial differences. 14. Figure 3 is too small that it will be difficult to understand. Why NDVI is much lower in case of coniferous forest? 15. Page 7 line 182. I think that it is not always so a late start of vegetation leads to a late end. 16. There is confusion in the section 3.2. Earlier the majority of the text was related to the use of NDVI data. But here VCI is analysed. I suggest the VCI should be more mentioned also in to abstract, introduction and methods description. 17. Page 11 lines246-247. Usually, the growing season is defined when the daily mean temperature is permanently higher $+5°C$. In that case, it starts much earlier than the end of April. In this study, an active growing season i.e. the $+10°C$ limit is used. These two terms should not be mixed. 18. I recommend writing of a separate section of conclusions where the main results of this study have been emphasized.

---

## Referee Comment (RC2) · Anonymous Referee #2 · 11 Apr 2017

This study investigates relationships between the different indices including NDVI, VCI, SPI1,3,6 and droughts over the east Baltic Sea region. The topic is relevant to ESD and special issue "Multiple drivers for Earth system changes in the Baltic Sea region". The paper is well-written and concise. I think the paper is ready to publish with only a few minor points to address. However, I have several minor comments 1. VCI analysis is a major part of the Result section. It is declared that it's better than NDVI. So abstract should include major results of VCI analysis. Methods of VCI assessment should be presented better 2. Section describing study area is needed. The first reason is that authors use datasets with different spatial coverage. The second is that study area is large enough and that's why climatic regime is different over different sub-regions. Instead, authors often provide point estimates (e.g. L126) 3. Separate section of conclusion is needed Detail comments and minor edits: P3L85 It's better

to use calendar dates P4L100 Interval assessments seem to be more appropriate because the region is large enough and there is a certain time lag in dates of natural phenomena between south-west and north-east regions. Figure 1 Maybe it's better to add some basic geographical information on the map: country names, main cities. "Sudy area" should be corrected P5L115 Abbreviation should be inputted after first mention. CORINE land cover (CLC) P5L116 It should be checked if year 2012 is covered by CLC 2000. Maybe CLC 2012? http://land.copernicus.eu/pan-european/corine-land-cover/clc-2012/view P5L120 Percentage of the joined types should be mentioned P5L125-129 Is it the temperature averaged over the study area or data from a certain station? Should be clarified P6L149 Maybe it's more useful to avoid weeks here and hereafter P8L184 Reference indicating that VCI is more suitable is needed P11L254 Shows, technical correction P12L285 I see no reason to compare this region with arid and semiarid areas

---

## Author Comment (AC1) · 8 May 2017

A. Does the title clearly reflect the contents of the paper? Partly. There is mentioned only NDVI but not VCI.

We believe the current title reflects the contents of paper. The VCI values were derived from initial NDVI data. In some cases, VCI values (with range from 0 to 100%) are more suitable for drought analysis but initial data set in any case is NDVI.

B. Does the abstract provide a concise and complete summary? Partly. Results from the section 3.2 were not presented.

We added information from the section 3.2 into abstract.

C. Is the overall presentation well structured and clear? The use of the VCI in not well

presented in the abstract, introduction and methods.

We added additional information about VCI in abstract, introduction and methods.

D. Is the language fluent and precise? Could be improved.

Language was improved.

Remarks and suggestions

1. The abstract is too general. I would like to see more concrete results of the study in the abstract.

We added additional information from the section 3.2 into abstract.

2. Page 1 line 29. There is written "Remote sensing of the vegetation condition is based on the fact that healthy plants have more chlorophyll and therefore absorbs more visible and reflects more infrared radiation (Myeni et al., 1995)". Plants reflect short-wave radiation and emit infrared radiation. I assume that here should be ". . . and emit more infrared radiation".

The line is correct as healthy plants absorb relatively more visible and reflect more near-infrared radiation, while damaged or sparse vege- tation tend to reflect more visible and absorb more near-infrared ra- diation. For clarity, the term "infrared" is changed to "near-infrared". https://earthobservatory.nasa.gov/Features/MeasuringVegetation/measuring_vegetation_2.php

3. Page 2 line 39. I suggest that vegetation response here not to climatic impacts but to different meteorological conditions each year.

We made correction. "It is necessary to emphasize that the vegetation (and hence NDVI values) response to the meteorological conditions in a given year depends on the geographical region...."

4. In fact, the NDVI describes the greenness of the underlying surface. Is the term

"greenness" used in this context and could it be used in this article?

NDVI describes how dense and green plant leaves are and have highest positive values in dense vegetation canopy (forests). NDVI reflects overall vegetative health. The term "greenness" is used and can be used in the studies involving NDVI, but we think in our study it is more appropriate to use term "vegetation condition" as we also apply VCI index. For this reason changes were made in the text.

5. I suggest more extended overview of the literature on climatological studies of droughts in the study region in the introduction.

The overview of the literature on climatological studies of droughts and drought effect on vegetation in the study region is included in manuscript.

6. Page 3 line 71. I suggest more exact definition of the study area. It does not cover the whole territory between 53-60 N and 20-30 E. Could it be identified as Estonia, Latvia, Lithuania and northeastern Poland? There is a contradiction between Figure 1 and other maps. In the first case, only these countries are shown on the map while in the second case, all land areas in the domain have been used. Can you explain this difference in the definition of the study region?

The description of spatial coverage of used data is added in text. All land areas in the domain have been used to present the results based on NDVI and consequently VCI. The CORINE data with 100 m resolution was used to identify the dominant land use in NDVI cells ($0,144° \times 0,144°$). CORINE covers only Estonia, Latvia, Lithuania and northeastern Poland in the study area. CORINE does not cover the rest of the area which is indicated as "No vegetation or land use data" in Figure 1.

7. Page 3 line 79. The end of the study period is not shown (from 1981 to the present). What is the last year?

2014. We made correction in the text.

8. Page 3 lines 85-86. Week numbers are usually not used in the everyday life. There-

fore, it is not informative to used week numbers from the beginning of a year. The use of dates is more useful in my mind.

The NOAA STAR–NESDIS NDVI data has weekly 7–day composite temporal resolution. We believe week numbers, as the original time format, is more appropriate to describe the missing values.

9. Page 4 Line 99. What is mean average daily air temperature? Is it simply daily mean temperature?

We made correction.

10. Page 4 pages 100-102. The beginning and the end of the active growing season varies significantly over the study region in dependence of latitude. It could be emphasize in the text. These dates are very variable also from year to year.

We added sentence: "It is necessary to mention that there is some territorial differences (due to latitude and distance from sea) as well as quite a big year-to-year variation of such dates".

11. Figure 1. It is very strange that the territory of Estonia and major part of Latvia is described only by broad-leaved forest. There are no cells from coniferous forest and arable land. This selection of cells does not represent adequately the vegetation types in that region. It should be explained somehow why these cells were selected. Here are also problems related to the CORINE land cover data. In the figure caption the CORINE data should be mentioned.

We mentioned CORINE in the caption of the figure. We used NDVI cells ($0,144° \times 0,144°$) only with stable land use (coincided at least in the 80 % of cell area over 1990 (CLC 1990) and 2012 (CLC 1990)) to reduce the uncertainties related to changes in land use cover. From the cells with stable land use the cells in which arable land, broad-leaved or coniferous forests were dominant (covers at least 50 % of the cell area) were selected. The selection of cells is described in the "Data and methods"

section. We agree that coniferous forests are common in Estonia and Latvia, but according to CORINE data they were not dominant in the NDVI cells or the land use has changed in 1990-2012 in the cells with large areas covered with coniferous forests. It is also possible, that large changes occurred not in the areas covered with coniferous forests, but in areas near forests, which were in the same cell.

12. It is not clear how the VCI values were calculated in this study. It should be described because they are used in this study.

The formula for VCI values calculation was presented in "Data and methods" section. However we added the additional information about VCI in this chapter.

13. Page 6 line 152. It is interesting to know when the week 25 takes place. I am not sure that the large values are more common in the northern part of the domain (Figure 2). Lower values are in the southern part but in the other regions there are not such territorial differences.

We added dates for week 25. We agree with reviewer interpretation (Figure 2) and made changes in the text.

14. Figure 3 is too small that it will be difficult to understand. Why NDVI is much lower in case of coniferous forest?

We increased the size of picture. The NDVI values depends on the amount of chlorophyll in the plants and on the structure of leaves. The evergreen coniferous forest have less chlorophyll than deciduous forest or grasslands, thus the NDVI value for pine forests are lower.

15. Page 7 line 182. I think that it is not always so a late start of vegetation leads to a late end.

We agree with referee. In the mentioned paragraph, we talked only about one particular year (1987). We extended the last sentence. "The late start of vegetation in 1987 also led to the late end but not in all years there is a close positive correlation between these

dates".

16. There is confusion in the section 3.2. Earlier the majority of the text was related to the use of NDVI data. But here VCI is analysed. I suggest the VCI should be more mentioned also in to abstract, introduction and methods description.

We made the suggested additions in abstract, introduction and data and methods.

17. Page 11 lines246-247. Usually, the growing season is defined when the daily mean temperature is permanently higher +5 C. In that case, it starts much earlier than the end of April. In this study, an active growing season i.e. the +10 C limit is used. These two terms should not be mixed.

We made corrections in the text.

18. I recommend writing of a separate section of conclusions where the main results of this study have been emphasized.

Conclusions were added.

―――――――――――――――――――

---

## Author Comment (AC2) · 8 May 2017

Several minor comments: 1. VCI analysis is a major part of the Result section. It is declared that it's better than NDVI. So abstract should include major results of VCI analysis. Methods of VCI assessment should be presented better.

We added some results of VCI analysis into abstract and extended the section of Data and methods with additional information about VCI.

2. Section describing study area is needed. The first reason is that authors use datasets with different spatial coverage. The second is that study area is large enough and that's why climatic regime is different over different sub-regions. Instead, authors often provide point estimates (e.g. L126)

The description of spatial coverage of used data is added in text. The CORINE data with 100 m resolution was used to identify the dominant land use in NDVI cells ($0,144° \times 0,144°$). CORINE covers only Estonia, Latvia, Lithuania and northeastern Poland in the study area. CORINE does not cover the rest of the area which is indicated as "No vegetation or land use data" in Figure 1. The short climatic and geographical description of the study area added to the introduction.

3. Separate section of conclusion is needed

Conclusions were added.

Detail comments and minor edits:

P3L85 It's better to use calendar dates

The NOAA STAR–NESDIS NDVI data has weekly 7–day composite temporal resolution. We believe week numbers, as the original time format, is more appropriate to describe the missing values.

P4L100 Interval assessments seem to be more appropriate because the region is large enough and there is a certain time lag in dates of natural phenomena between southwest and north-east regions.

We have provided additional information in the text.

Figure 1 Maybe it's better to add some basic geographical information on the map: country names, main cities. We added the labels for countries and the Baltic Sea.

P5L115 Abbreviation should be inputted after first mention. CORINE land cover (CLC) P5L116 It should be checked if year 2012 is covered by CLC 2000. Maybe CLC 2012? http://land.copernicus.eu/pan-european/corineland-cover/clc-2012/view

We have provided abbreviation in the text and corrected CLC 2000 to CLC2012.

P5L120 Percentage of the joined types should be mentioned.

We added the description of the average composition of joined land use type in the text.

P5L125-129 Is it the temperature averaged over the study area or data from a certain station? Should be clarified

The answer is "averaged over the study area". We made clarification in the text.

P6L149 Maybe it's more useful to avoid weeks here and hereafter

We substituted the week numbers with dates or indicated the beginning or the end of month if indicating of the exact date is not feasible.

P8L184 Reference indicating that VCI is more suitable is needed

We added the reference (Jain et al., 2010)

P11L254 Shows, technical correction

We corrected the error.

P12L285 I see no reason to compare this region with arid and semiarid areas.

We try to examine the feasibility of NDVI to identify the drought effect on vegetation in the eastern part of the Baltic Sea region. We compare this region with arid and semiarid areas to describe the baseline for NDVI application. In dry areas vegetation is primarily related to precipitation patterns and NDVI is usually a good indicator of droughts. In the eastern Baltic Sea region there are at least several significant drivers for vegetation vigor.
* * *